# Protective Potential and Functional Role of Antibodies Against SARS-CoV-2 Nucleocapsid Protein

**DOI:** 10.3390/antib14020045

**Published:** 2025-05-28

**Authors:** Alexandra Rak, Ekaterina Bazhenova, Polina Prokopenko, Victoria Matyushenko, Yana Orshanskaya, Konstantin V. Sivak, Arina Kostromitina, Larisa Rudenko, Irina Isakova-Sivak

**Affiliations:** 1Institute of Experimental Medicine, St. Petersburg 197022, Russiamatyshenko@iemspb.ru (V.M.); isakova.sivak@iemspb.ru (I.I.-S.); 2Smorodintsev Research Institute of Influenza, St. Petersburg 197376, Russiakonstantin.sivak@influenza.spb.ru (K.V.S.)

**Keywords:** COVID-19 vaccine, SARS-CoV-2, Syrian hamster, nucleocapsid protein, antiviral antibodies

## Abstract

Cases of new COVID-19 infection, which manifested in 2019 and caused a global socioeconomic crisis, still continue to be registered worldwide. The high mutational activity of SARS-CoV-2 leads to the emergence of new antigenic variants of the virus, which significantly reduces the effectiveness of COVID-19 vaccines, as well as the sensitivity of diagnostic test systems based on variable viral antigens. These problems may be solved by focusing on highly conserved coronavirus antigens, for example nucleocapsid (N) protein, which is actively expressed by coronavirus-infected cells and serves as a target for the production of virus-specific antibodies and T cell responses. It is known that anti-N antibodies are non-neutralizing, but their protective potential and functional activity are not sufficiently studied. Here, the protective effect of anti-N antibodies was studied in Syrian hamsters passively immunized with polyclonal sera raised to N(B.1) recombinant protein. The animals were infected with 10^5^ or 10^4^ TCID_50_ of SARS-CoV-2 (B.1, Wuhan or BA.2.86.1.1.18, Omicron) 6 h after serum passive transfer, and protection was assessed by weight loss, clinical manifestation of disease, viral titers in the respiratory tract, as well as by the histopathological evaluation of lung tissues. The functional activity of anti-N(B.1) antibodies was evaluated by complement-dependent cytotoxicity (CDC) and antibody-dependent cytotoxicity (ADCC) assays. The protection of anti-N antibodies was evident only against a lower dose of SARS-CoV-2 (B.1) challenge, whereas almost no protection was revealed against BA.2.86.1.1.18 variant. Anti-N(B.1) monoclonal antibodies were able to stimulate both CDC and ADCC. Thus, anti-N(B.1) antibodies possess protective activity against homologous challenge infection, which is possibly mediated by innate Fc-mediated immune reactions. These data may be informative for the development of N-based broadly protective COVID-19 vaccines.

## 1. Introduction

A novel COVID-19 infection caused by SARS-CoV-2 continues to pose severe socioeconomic burden in all countries of the world. Prevention of infection with new antigenic variants of SARS-CoV-2 using licensed vaccines and antiviral drugs based on monoclonal antibodies turned out to be ineffective due to the constant evolutionary variability of the virus [1]. In this regard, an urgent task of public healthcare today is the development of new cross-protective vaccines based on highly conserved SARS-CoV-2 proteins, the protective effects of which do not depend on the variability of the antigenic properties of the virus. One such promising antigens is the nucleocapsid protein (N), which is significantly less variable than the surface-exposed spike protein (S) [2,3]. According to the literature data, N protein is actively produced in infected cells, including being detected on the cell surface during the progression of natural infection [4] and is highly immunogenic, inducing the formation of specific T cells and antibodies [5]. The vast majority of studies consider N protein to be a target for the induction of T cell immune responses [6,7,8], whereas no convincing data on the ability of anti-N antibodies to protect against natural infection have been presented so far.

Although antibodies against N protein lack neutralizing activity [9], they are intensively produced [10], persist in the organism significantly longer than S-specific antibodies [11], and may hypothetically provide antiviral protection by their functional activity, acting as activators of complement cascade reactions [12] or antibody-mediated cellular phagocytosis/cytotoxicity (ADCP/ADCC) [13]. In addition to natural infection, the production of N-specific antibodies can also be stimulated by immunization with whole-virion inactivated [14,15] or live attenuated [16] vaccines, but the mechanism of participation of these immunoglobulins in protection against COVID-19 remains largely unexplored. Moreover, there is evidence that antibodies to N protein may have autoimmune properties, provoking the development of various immunopathologic conditions [17]. 

Studying the protective role of anti-N antibodies is necessary both in light of the prediction of infection course/outcomes and for the development of universal N-based vaccines against COVID-19. Here, we evaluated the antiviral potential of anti-N immunoglobulins against homologous and heterologous infections and studied a possible mechanism of this protection.

## 2. Materials and Methods

### 2.1. Cells, Virus, N(B.1) Protein, Anti-N(B.1) Antibodies, and Sera

African green monkey kidney Vero CCL81 cells were purchased from the American Type Culture Collection (ATCC) and maintained in DMEM supplemented with 10% fetal bovine serum (FBS) and 1 × antibiotic–antimycotic (AA) (all from Capricorn Scientific, Ebsdorfergrund, Germany). SARS-CoV-2 viruses HCoV-19/Russia/StPetersburg-3524/2020 (B.1 Lineage, Wuhan) and hCoV-19/Russia/SPE-RII-9293S/2023 (BA.2.86.1.1.18 Lineage, Omicron) were obtained from the Smorodintsev Research Institute of Influenza (Saint Petersburg, Russia). They were propagated and titrated as previously described [18] in Vero CCL81 or Vero E6 cells, respectively, at MOI 0.01 using DMEM supplemented with 2% FBS, 10 mM of HEPES, and 1 × AA (all from Capricorn Scientific, Ebsdorfergrund, Germany) at 37 °C and 5% CO_2_. As full cytopathic effect was achieved, the virus-containing media were harvested, clarified via centrifugation at 3000 rpm, and stored at −70 °C in aliquots. All experiments with live SARS-CoV-2 were performed in a biosafety level-3 (BSL-3) laboratory.

N(B.1) his-tagged protein of bacterial origin was obtained from *E. coli* BL21(DE3) strain-producer cells by 0.5 mM IPTG induction performed at +37 °C for 4 h and followed by immobilized metal affinity chromatography accordingly to the previously described optimized protocol [19]. 

Anti-N(B.1) mouse monoclonal antibodies NCL2, NCL5, NCl7, and NCL10 were previously purified from hybridoma-induced ascitic fluids by protein A chromatography, as described in [20].

Neutralizing rabbit polyclonal anti-RBD antibody was purchased from JSC BIOCAD, Russia.

This study used naïve hamster sera as well as sera obtained from golden Syrian hamsters intraperitoneally immunized three times with 14-day intervals with 100 µg of recombinant N(B.1) protein.

### 2.2. Hamster Study Design

Golden Syrian hamsters (*Mesocricetus auratus*, females 4–8 weeks old) were purchased from Stezar cattery (Vladimir region, Russia). The experimental design was approved by the local Ethics Committee of the Institute of Experimental Medicine (protocol no 4/24, dated 24 October 2024). The animals were maintained in standard laboratory vivarium conditions with free access to food and water. This study was conducted in accordance with Directive 2010/63/EU [21].

Animals aged 8 weeks and body weight 80–100 g (five animals per group as minimally required for proper statistical assessment; twenty hamsters in total) were pre-screened in ELISA on 2 µg/mL immobilized recombinant N(B.1) protein and then intraperitoneally immunized with 2 mL of anti-N(B.1) or control hamster serum previously inactivated for 1 h at 56 °C and 1:1 diluted with PBS (n = 10 for each serum). This dosage was chosen as an optimal given the safe administration volume and titer of anti-N(B.1) sera obtained. Then, 6 h later, each of the two groups was randomly divided into two challenge subgroups (n = 5) and intranasally infected with HCoV-19/Russia/StPetersburg-3524/2020 (B.1, Wuhan) or hCoV-19/Russia/SPE-RII-9293S/2023 (BA.2.86.1.1.18, Omicron) viruses at a dose of 10^4^ or 10^5^ TCID_50_ under light ether anesthesia. Clinical symptoms of the disease and body weight dynamics were monitored for 6 days after the challenge. The clinical features of the disease course were evaluated according to the following criteria: coat condition: 0—normal, 1—lack of care; interaction with other animals: 0—normal, 1—reduced; feed consumption: 0—normal, 1—reduced; behavior in the open area: 0—active, 1—reduced; reaction to being taken: 0—normal, 1—reduced. The inoculations and measurements were performed in strict order, the animals were marked with 0.5% picric acid, and cages were placed in different locations to minimize potential confounders. The qualified personnel were aware of the group allocation at all stages of the experiment.

On the 6th day, to assess the tissue levels of virus replication, animals were humanely removed from the experiment by ether overdose, and nasal passages and lungs were aseptically isolated. The tissues were weighed and homogenized with the steel beads in the Qiagen TissueLyser LT, followed by titration on Vero CCL81 (for B.1 virus) or Vero E6 cells (for BA.2.86.1.1.18 virus) with FFU counting by the previously described approach [22]. For this, 10^−1^ to 10^−3^ dilutions of clarified homogenates were used to infect 96-well cell monolayers in triplicate. After 20 h of incubation, virus-infected cells were formalin-fixed overnight (ON), permeabilized, and then consistently treated with anti-N(B.1) biotynilated antibody and eGFP–streptavidin fusion protein. FFU counting was performed using AID vSpot Spectrum (Autoimmun Diagnostika GmbH, Strassberg, Germany).

One lung lobe was intended for histological studies and fixed in 10% buffered formalin solution (pH = 7.4) for 48 h. Then, routine histologic wiring was applied on a histoprocessor Histo-Tek VP1 (Sakura, Japan), followed by embedding of the specimens in paraffin blocks. The 3 μm slices were stained with hematoxylin and eosin solution according to a standard protocol or by the Martius Scarlet Blue (MSB) stain for elective visualization for blood clots and vascular condition. In the last case, fibrin blood clots/red blood cells, adventitial collagen, and lymphocytes are dyed red, blue, and purple, respectively. The stained slides were analyzed using the LEICA DM1000 microscope, while capturing and measurements were performed using the ADF Image Capture 4.17 software package. Morphometric analyses were performed as previously described [23], calculating alveolar septa thickness and semi-quantitatively scoring the degree of involvement of the airways, pulmonary parenchyma, and vasculature separately to evaluate the damage to lung tissues.

### 2.3. Assessment of Functional Activity of Anti-N Antibodies

The ability of anti-N(B.1) antibodies to promote Fc-mediated innate immune reactions was assessed both by the antibody-dependent natural killer (NK) cell degranulation assay, which is a surrogate assay for the assessment of antibody-dependent cellular cytotoxicity (ADCC) responses [24], and by the complement-dependent cytotoxicity assay (CDC). In the first case, 2 µg/mL of recombinant N(B.1) protein in carbonate–bicarbonate buffer was used to coat high-sorbent 96-well plates (Corning, NY, USA) at 4 °C overnight. Then, anti-N(B.1) mAbs (5 µg/mL) or serum samples (1:50) were added to the washed wells and incubated at 37 °C for 1 h. Then, 2 × 10^6^ splenocytes collected from naïve C57BL/6J mice were added in 100 µL of CR-10 to each well and incubated at 37 °C and 5% CO_2_ for 24 h. On the next day, the cell suspensions were collected and treated with ZombieAqua fixable viability dye, anti-CD3 (clone 2E7), anti-CD49b (clone DX5), anti-CD45.2 (clone 104), and anti-CD107a (clone 1D4B) conjugated antibodies (Biolegend, San Diego, CA, USA) diluted in staining buffer (SB) (PBS supplemented with 0.2% BSA and 0.05% sodium azide) for 20 min at RT in the dark. In this staining protocol, a transmembrane protein CD107a of cytolytic granules was targeted on the cell surface as a degranulation marker [25]. The staining procedure was followed by washing with 200 µL of SB twice, fixation of the cells in 1% formaldehyde, and flow cytometry analysis. At least 30,000 events were measured using a CytoFlex Cytometer equipped with CytExpert v 2.4.0.28 software (Beckman Coulter, Brea, CA, USA).

To perform the CDC assay, Vero CCL81 monolayers seeded the day before on 24-well plates were infected with an HCoV-19/Russia/StPetersburg-3524/2020 (B.1 Lineage, Wuhan) virus at MOI 0.01 and incubated in DMEM supplemented with 2% FBS and 1 x AA at 37 °C in 5% CO_2_ overnight. Then, the medium was removed and the cells were washed with 300 µL of PBS, followed by treatment with 10 µg of anti-N(B.1) mAbs or 1:2 sera diluted in DMEM at 37 °C and 5% CO_2_ for 15 min. Then, 50 µL of naïve guinea pig sera diluted 1:10 in DMEM was added as a complement source, and the plates were further incubated for 3 h at 37 °C and 5% CO_2_. The well supernatants were transferred to the 2 mL tubes, and the remaining cell monolayers were washed twice with PBS prior to the accutase treatment for 10 min at 37 °C and 5% CO_2_. After the accutase inactivation by the addition of 1 mL of DMEM supplemented with 10% FBS, the cells were thoroughly resuspended and collected to the same tubes with the corresponding supernatants. The tubes were centrifuged for 7 min, 300× *g*, washed with PBS twice, resuspended in 300 µL of PBS, stained with propidium iodide and YO-PRO iodide (Thermo Fisher Scientific, Waltham, MA, USA), and analyzed using the CytoFlex Cytometer. CDC induction was determined as an increase in the percent of cells in the late apoptotic phase.

### 2.4. Statistical Analysis

Data were processed using the statistical tool of GraphPad Prism 6.0 Software (GraphPad Software, San Diego, CA, USA). Compliance with the normal distribution was checked by the Shapiro–Wilk test. Differences between several test groups were analyzed by two-way ANOVA with Sidak’s correction. Pairwise comparisons between two groups were performed by the nonparametric Mann–Whitney *t*-test. The significance level was set at *p* < 0.05.

## 3. Results

### 3.1. Protection Against SARS-CoV-2 Infection

For this study, we generated hyperimmune hamster sera against recombinant N(B.1) protein for subsequent use for serum passive transfer experiments. To collect it, we expressed the antigen in bacterial system (*E. coli* BL21(DE3) cells) using a previously developed protocol [5] and obtained sufficient quantities for triple immunization of five hamsters with 100 µg of recombinant N(B.1) protein with a 14-day interval. The resulting titer of pooled anti-N(B.1) sera assessed in ELISA on N(B.1) was 1:7290. At the same time, the titer of naïve sera was also checked and appeared to be lower than the ELISA sensitivity threshold.

The pre-screened naïve hamsters were immunized intraperitoneally with 2 mL of 0.22 µm filtered naïve or anti-N(B.1) sera diluted 1:1 in PBS and 6 h later challenged with 10^4^ or 10^5^ TCID_50_ of SARS-CoV-2 B.1 (Wuhan) or BA.2.86.1.1.18 (Omicron). The animals were weighed and monitored prior to the infection and then daily. The nasal washes were also collected just before the challenge (day 0) and then on day 2 and 4.

On the 6th day after challenge with Wuhan (B.1) variant, lungs and nasal turbinates were harvested from immunized and control hamsters, where SARS-CoV-2 (B.1) titers were determined by FFU counting in Vero CCL81. Surprisingly, anti-N(B.1) sera significantly reduced virus replication both in the upper and lower respiratory tract if the challenge dose was 4 lg, while in the case of 5 lg reduction in nasal viral titers compared to the mock group it did not reach statistical significance (Figure 1a,b). At the same time, no significant reduction in viral titers in the nasal washes of the immunized animals was revealed (Figure 1c), which may be associated with the non-mucosal nature and intraperitoneal administration method of serum anti-N(B.1) immunoglobulins. Taken together, the data obtained confirm the ability of anti-N antibodies to form protective immunity against the homologous challenge virus at a low dose in the upper and lower respiratory tract.

Body weight and wellness score monitoring conducted within 6 days after the challenge revealed that anti-N(B.1)-immunized hamsters challenged with 4 lg of B.1 (Wuhan) were protected from weight loss from the 4th day of infection, as revealed by the analysis of weight dynamics (Figure 2a), while area under the curve (AUC) weight loss values were statistically insignificant (Figure 2b). At the same time, immunized animals were less affected by the infection caused by 4 lg of B.1 (Wuhan) than animals that received the control sera as they were protected from progression of the clinical signs of the infection (Figure 2c), while this effect was not observed at a higher challenge dose, probably because of the insufficient protective effect of anti-N(B.1) antibodies against the more faster viral propagation. This may be a consequence of differences in the pathogenesis of infections caused by two doses of virus, resulting in different kinetics of titer accumulation. These assumptions were further confirmed by the histopathologic and morphometric examination of lung tissue, showing different patterns of infections caused by 4 and 5 lg of viruses, despite similar pulmonary titers being determined (Figure 1b).

In contrast to the results of the B.1 challenge study, anti-N(B.1) sera appeared to be ineffective against the infection caused by both doses of BA.2.86.1.1.18 strain without affecting either viral titers in respiratory organs and nasal washes (Figure 3) or pronounced protection of the well-being of animals during the challenge study, although a weak difference in the AUC values of weight loss was detected (Figure 4).

The limited protective efficacy of anti-N(B.1) antibodies, revealed only against homologous challenge virus, is in good agreement with our previous findings of slow cumulating changes in the antigenic properties of the N protein during SARS-CoV-2 evolution [10.3390/v15010230], and thus the inability of anti-N(B.1) sera to prevent the progression of infection caused by an evolutionarily novel SARS-CoV-2 variant may be a consequence of the appearance of the adaptive mutations in the B-cell epitopes of the N antigen.

These data are in line with the histopathology assessment of the hematoxylin and eosin (H&E) staining of lung tissues collected from the SARS-CoV-2 (B.1)-challenged hamsters, which revealed that approximately 60% of the histological slides in the control group were characterized by more than 50% damage to lung and vascular tissues. Furthermore, this group had the most severe alveolar inflammation with loss of normal septal histoarchitectonics (Figure 5, left panels). In addition, there was a more frequent desquamation of the necrotic bronchiolar epithelium into the bronchi lumen compared with the other control groups. In the group immunized with anti-N(B.1) sera and challenged with 4 lg of B.1 virus, lung pathology was less pronounced compared to the control animals. In particular, alveoli were affected in a smaller percentage and at a lower severity. Bronchioles as well as vessels were affected mainly in the apical segments of the lungs. In animals which received anti-N(B.1) sera and 5 lg of SARS-CoV-2 (B.1), the lung histoarchitectonics were not comparable to those of the intact hamsters’ lungs, but this group was the least affected by B.1 (Wuhan) challenge.

MSB staining was alternatively used for elective staining for blood clots and assessment of vascular condition (Figure 5, right panels). In this case, the fibrin blood clots and red blood cells are dyed in red, while collagen adventitia is in blue and lymphocytes are in purple. This staining approach allowed us to reveal that the condition of the lung parenchyma in naïve animals had no features, which corresponded to the species norm of hamsters. There were no blood clots in the lumens of large vessels, and the wall thickness of the vessels was normal. At the same time, in the C-sera-immunized groups infected with 4 or 5 lg of B.1 (Wuhan) strain, we detected a decrease in the airiness of the pulmonary parenchyma due to thickening of the alveolar septa and vascular damage, the presence of intraluminal thrombi, edema and thickening of the media, and edema of the adventitia with progressed infiltration of lymphocytes (Figure 5a). The administration of anti-N(B.1) sera, especially in the group infected with 4 lg of the B.1 (Wuhan) virus, resulted in protective effects such as an increase in the airiness of the lung parenchyma (by 40%) due to the lower thickness of the alveolar septa, as well as protection against vascular damage. In this group, the magnitude of perivasculitis was moderate compared to that of the infection control. In the case of challenge with 5 lg of the B.1 (Wuhan) virus, the effect of immunization was not so significant.

In a series of experiments with infection with the BA.2.86.1.1.18 (Omicron) strain (Figure 5b), the alveolar septa were less damaged in the hamsters immunized with C-sera, but there was a greater lesion of the perivascular space due to lymphocytic infiltration. It should be noted that the use of anti-N(B.1) sera led to an apparent increase in vascular damage and the development of perivasculitis.

We further calculated and compared the thickness of the alveolar septa in hamsters immunized with C- or anti-N(B.1) sera and challenged with Wuhan (B.1) or Omicron (BA.2.86.1.1.18) viruses (Figure 6). It was found that regardless of the infecting strain, the values were significantly closer to those in the naïve animals in hamsters vaccinated with anti-N(B.1) sera, and thickness differences between the animals which received C- or anti-N(B.1) sera were more pronounced in the Wuhan (B.1) challenge study (Figure 6a).

Semi-quantitative assessment of pulmonary/alveolar pathology (Figure 7) showed significant differences in the degree of involvement of bronchi and bronchioles in the pathological process (Figure 7a), as well as in the severity of pulmonary/alveolar pathology (Figure 7b) and vascular lesions (Figure 7c) in animals from groups immunized with control versus anti-N(B.1) sera and infected with 4 lg of B.1 virus. At the same time, no protective effects of anti-N(B.1) sera against BA.2.86.1.1.18 challenge virus were found (Figure 7d–f).

### 3.2. Functional Activity of the N(B.1)-Specific Antibodies

In our previous experiments, no direct inhibition of SARS-CoV-2 propagation in Vero CCL81 by anti-N sera or monoclonal antibodies was revealed (Figure 8). Here, rabbit neutralizing polyclonal antibody against the receptor binding domain (RBD) of the spike(B.1) protein was used as a positive control. This findings are in line with the well-known data on the non-neutralizing properties of anti-N antibodies [9].

As the direct neutralization of the virus appeared to be ineffective, we further attempted to elucidate the mechanisms by which the defense by anti-N(B.1) antibodies is mediated. Due to the non-neutralizing nature and the obvious impact of the anti-N antibodies on the SARS-CoV-2-induced cross-protective potential, it is important to assess the functional activity of these antibodies, i.e., Fc-driven cytotoxicity effects. The functional activity of the anti-N(B.1) sera and monoclonal antibodies was assessed in two assays, complement-dependent cytotoxicity and an antibody-dependent natural killer (NK) degranulation activity, which served as a surrogate assay for the evaluation of antibody-dependent cellular cytotoxicity (ADCC).

The specificity of anti-N(B.1) monoclonal antibodies NCL2, NCL5, NCL7, and NCL10 has been studied previously [20]. Although these antibodies were produced against the same recombinant N(B.1) antigen, they have distinct isotypes and epitope specificity, which may account for differences in their properties.

#### 3.2.1. Complement-Dependent Cytotoxicity (CDC)

The antiviral defense mechanisms that N-protein-based vaccines are aimed to trigger have not yet been definitively established. One of the actual questions is whether the complement system plays an important role in this protection. A previously obtained serum sample from COVID-19 convalescent and sera from mice immunized three times with recombinant N(B.1) protein, as well as anti-N monoclonal antibodies [20], were added to Vero CCL81 cells infected with HcoV-19/Russia/StPetersburg-3524/2020 (B.1 Lineage, Wuhan) virus, followed by incubation with guinea pig naïve sera (the complement source); the cell control wells contained no antibodies or sera. This resulted in the increased late apoptosis of the infected cells in the case of treatment with convalescent serum and NCL10 antibody (Figure 9a), possibly as a consequence of the unique properties of the latter. The assay served as an indicator which is considered to correlate well with the CDC-mediating antibody activity [26]. 

Interestingly, serum anti-N(B.1) antibodies from immunized mice demonstrated no CDC activity, compared to monoclonal antibodies NCL10, and in the last case, the percentage of induced late apoptotic cells was comparable to those increased by the addition of convalescent sera. Since the hamsters passively immunized with anti-N(B.1) sera had significantly reduced viral pulmonary titers six days after the challenge compared to the control group, the promotion of CDC response by the convalescent sera and anti-N monoclonal antibodies suggests that the complement system may play an important role in reducing SARS-CoV-2 propagation in the lungs.

#### 3.2.2. Antibody-Dependent Cellular Cytotoxicity (ADCC)

Due to the lack of fluorescently labeled antibodies to the surface markers of Syrian hamsters, the possible ADCC activity of N-specific antibodies was assessed for the previously generated mouse monoclonal antibodies NCL2, NCL5, NCL7, and NCL10. A stimulation of antibody-dependent cellular cytotoxicity was revealed for two of the four mAbs, NCL5 and NCL10, as evidenced by the level of NK cell degranulation induced by incubation of these antibodies with splenocytes of naïve C57BL/6J mice as a source of NK cells (Figure 10). Here, phorbol 12-myristate 13-acetate (PMA) was used as a positive degranulation control, while no sera were added to the cell control wells. These results suggest that the ADCC mechanism may play some role in the protection from SARS-CoV-2 infection provided by anti-N antibodies, although these findings are not directly transferable to the results of evaluating the protective activity of antibodies in hamster experiments.

## 4. Discussion

Despite the official end of the COVID-19 pandemic declared by the WHO in May 2023 [27], this disease continues to cause significant socioeconomic damage to the global community. Effective control of the spread of this infection is impossible without understanding the molecular mechanisms of its progression, and in the absence of highly specific etiotropic therapeutics and vaccine prophylaxis for COVID-19. Although the use of first-generation vaccines was able to contain the spread of the original SARS-CoV-2 strain and minimize the impact of the early phase of the COVID-19 pandemic [28,29], they predominantly targeted the coronavirus spike protein, which has led to the global spread of SARS-CoV-2 antigenic variants and a significant reduction in vaccine efficacy [30,31].

Due to the considerable variability of coronavirus surface antigens, the efforts of researchers have been focused on creating universal vaccines against COVID-19 based on the most conserved components of SARS-CoV-2 [32]. One of them is the N protein of coronavirus, the main component of virions actively synthesizing by infected cells and causing the intensive production of antiviral antibodies [33]. The N antigen also serves as one of the main targets of antiviral T cell immune responses [8], which makes it a promising basis for the development of universal recombinant vaccines against COVID-19 [2,3]. 

Despite the general characterization of the properties of anti-N antibodies produced in response to immunization or during coronavirus infection, their protective activity still remains poorly understood. Some early studies described the ability of monoclonal antibodies specific to the N protein of murine hepatitis coronavirus to protect animals from wild-type challenge [34]. In addition, the efficacy of prior passive immunization with monoclonal anti-N antibodies or serum from animals immunized with the N protein of SARS-CoV-2 to mitigate the effects of infection was previously demonstrated by Dangi et al. [35]. N-specific antibodies are known to be non-neutralizing [9], and the ability to trigger ADCC has been shown, including for antibodies against SARS-CoV-2 that do not have neutralizing activity [36,37,38]. Moreover, there is evidence for the preferential activation of NK cells by antibodies against SARS-CoV-2 antigens other than the spike protein [39]. Antibodies generated in response to COVID-19 infection were found to be able to induce ADCC significantly more actively compared to antibodies generated in response to vaccination [40], and this fact also indicates a possible role for anti-N antibodies in mediating innate cell protective responses. The direct possibility of the induction of ADCC and ADCP reactions by anti-N antibodies from the plasma of COVID-19 convalescents was recently shown by Díez et al. [41] and Hagemann et al. [42]. In addition, the ability of some monoclonal N-specific antibodies against SARS-CoV-2 [12] and anti-N antibodies generated in COVID-19 convalescents [43] to induce CDC initiation has been demonstrated. Here, we also found that anti-N(B.1) monoclonal antibodies were able to provoke CDC, and some of them had an ADCC-stimulating ability.

Moreover, our results clearly indicate the protective properties of anti-N(B.1) antibodies against homologous challenge, and the inability of sera against N(B.1) to inhibit the infection caused by the evolutionary distant BA.2.86.1.1.18 variant, which is in line with the previously obtained data on the restricted specificity of anti-N(B.1) sera to the recombinant N protein of more recent variants of the coronavirus [20]. The combination of these facts suggests that antiviral antibody generation alone is insufficient for protection and that the induction of T cell immune responses is necessary for the elimination of infection.

The fact of the involvement of N-specific antibodies in the realization of innate immunity reactions makes the development of N-based vaccines very promising, but a careful study is required in light of the reported autoreactivity of antiviral antibodies generated in COVID-19 convalescents [44,45] and, in particular, of anti-N immunoglobulins, especially in light of the predicted similarity of low-complexity regions of N protein and components of the human proteome [46]. The evaluation of self-reactive properties of anti-N antibodies is mandatory to assess the risk of autoimmune pathologies, which may be due to their partial specificity to the human antigens and should be considered an undesirable effect of their generation caused by vaccination or natural infection [17,44]. Indeed, anti-SARS-CoV-2 antibodies produced in pediatric patients with multisystem inflammatory syndrome in children (MIS-C) have been shown to exhibit increased reactivity to a distinct domain of the N protein that shares remarkable structural similarity with the human antiviral response regulator SNX8, showing an example of molecular mimicry to avoid host defense by the virus [47]. Moreover, it is suspected that anti-N antibodies produced during SARS-CoV-2 re-infection or immunization with inactivated whole-virion vaccines may provoke the antibody-dependent enhancement of infection (ADE) and the development of long COVID syndrome [48] by counteracting the neutralizing anti-spike immunoglobulins and intensifying the production of pro-inflammatory cytokines, resulting in a cytokine storm and multiple organ failure [49]. In particular, it has been shown that mouse monoclonal anti-N isotype IgG1 antibodies in the presence of N protein are able to stimulate IL-6 production by myeloma-derived K-ML2 cells [50] and in macrophage cultures [51]. In vivo experiments with the related virus SARS-CoV have shown that pre-vaccination of mice by recombinant vaccinia virus (VV) that expressed N protein during subsequent challenge resulted in the active production of Th1/Th2 pro-inflammatory cytokines (IFN-γ, IL-2, IL-4, IL-5) in animals and the development of severe pneumonia [52]. Taken together, these data indicate that an excessive host immune response against the N protein, including the formation of self-reactive antibodies [53], may be involved in disease progression. In light of innate protection mechanisms triggered by N protein, further studies should be focused on evaluating the feasibility of vaccination that induces the production of anti-N antibodies versus the risk of developing autoimmune pathologies.

## 5. Conclusions

Our findings suggest that anti-N antibodies, while not possessing direct neutralizing activity, can provide protection against homologous SARS-CoV-2 infection, presumably by activating innate immunity responses such as CDC and ADCC reactions. The obtained results can be used to predict the possible consequences of coronaviral infection, as well as for the further development of N-based vaccines against COVID-19. Further studies are needed to clarify the type of triggered defense reactions and to elucidate the mechanisms of Fc-effector activation.

## Figures and Tables

**Figure 1 antibodies-14-00045-f001:**
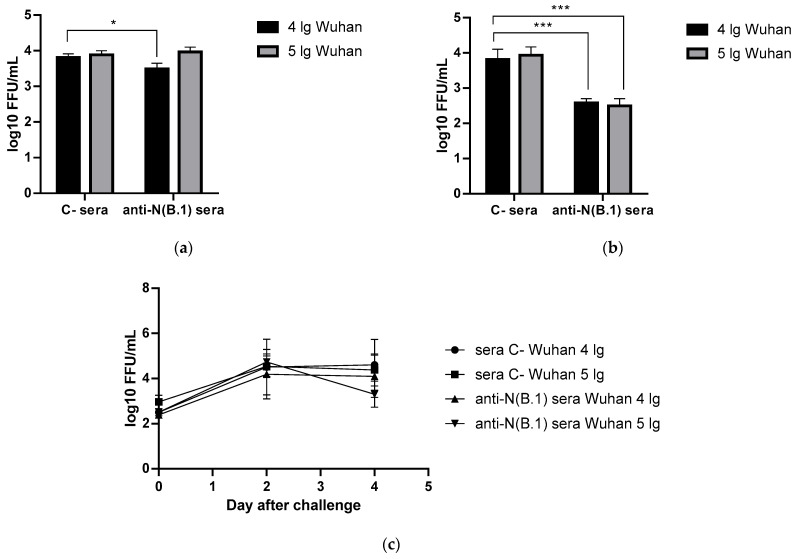
Viral replication rates in animals immunized with control or anti-N(B.1) sera and challenged with Wuhan (B.1) virus (n = 5 for each challenge dose): (**a**) in nasal turbinates; (**b**) in lung tissue; (**c**) in nasal washes. *—*p* < 0.05; ***—*p* < 0.001.

**Figure 2 antibodies-14-00045-f002:**
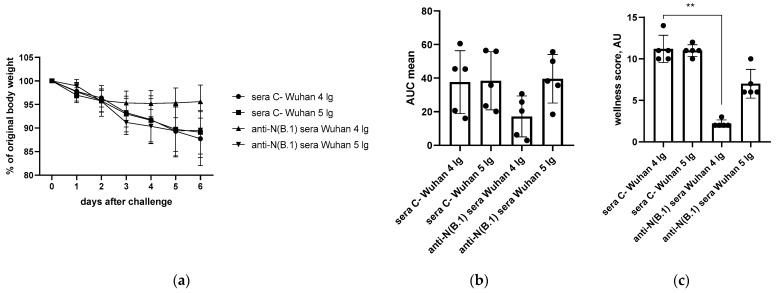
The monitoring of performance of hamsters immunized with anti-N(B.1) or control sera after B.1 (Wuhan) challenge (n = 5 for each challenge dose): (**a**) individual body weight dynamics; (**b**) areas under the curves (AUCs) of weight loss calculated as a trapezoidal square for each hamster and expressed in arbitrary units; (**c**) cumulative mean wellness score at day 6 after the challenge. **—*p* < 0.01.

**Figure 3 antibodies-14-00045-f003:**
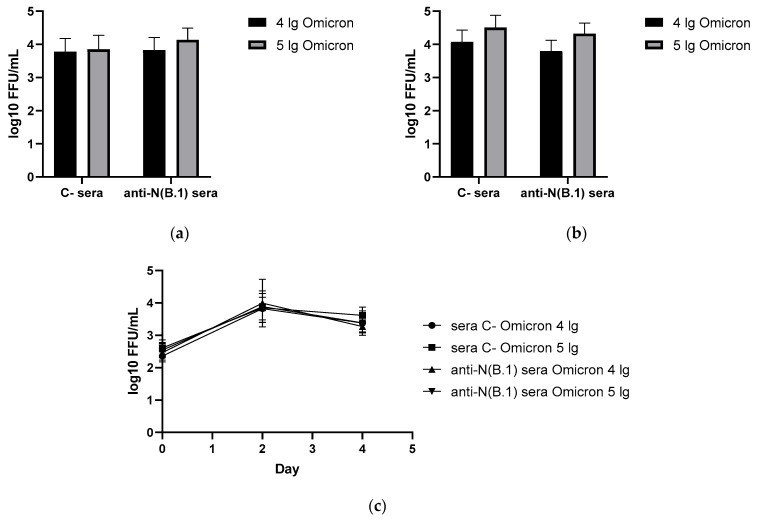
Viral replication rates in animals immunized with control or anti-N(B.1) sera and challenged with Omicron (BA.2.86.1.1.18) virus (n = 5 for each challenge dose): (**a**) in nasal turbinates; (**b**) in lung tissue; (**c**) in nasal washes.

**Figure 4 antibodies-14-00045-f004:**
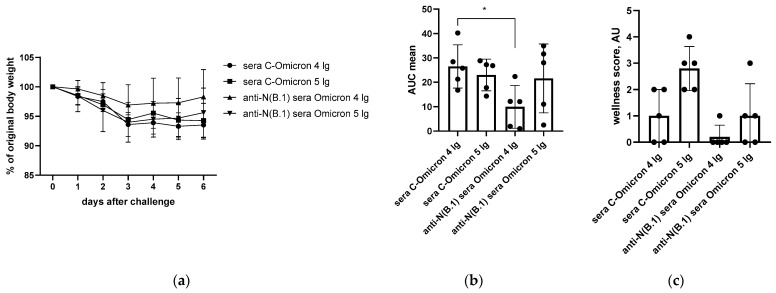
The monitoring of performance of hamsters immunized with anti-N(B.1) or control sera after Omicron (BA.2.86.1.1.18) challenge (n = 5 for each challenge dose): (**a**) individual body weight dynamics; (**b**) areas under the curves (AUCs) of weight loss calculated as a trapezoidal square for each hamster and expressed in arbitrary units; (**c**) grouped wellness score. *—*p* < 0.05.

**Figure 5 antibodies-14-00045-f005:**
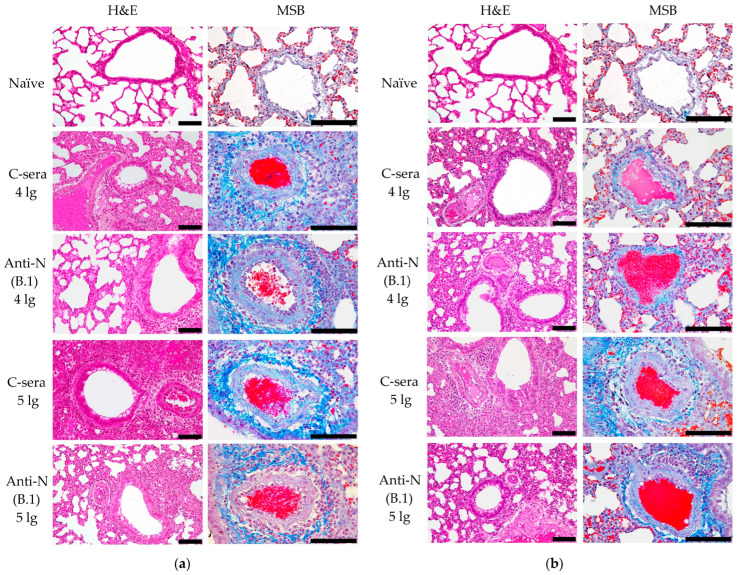
Histopathological assessment of protective effect of anti-N(B.1) sera against Wuhan (B.1) (**a**) and Omicron (BA.2.86.1.1.18) (**b**) viruses in a hamster model on day 6 after challenge. Representative micrographs of hematoxylin–eosin or MSB-stained lung sections are shown at 200× and 400× magnifications, respectively. The identical microphotographs of lung tissue from the naïve animals were used for two panels (first row). Scale bars refer to 100 µm.

**Figure 6 antibodies-14-00045-f006:**
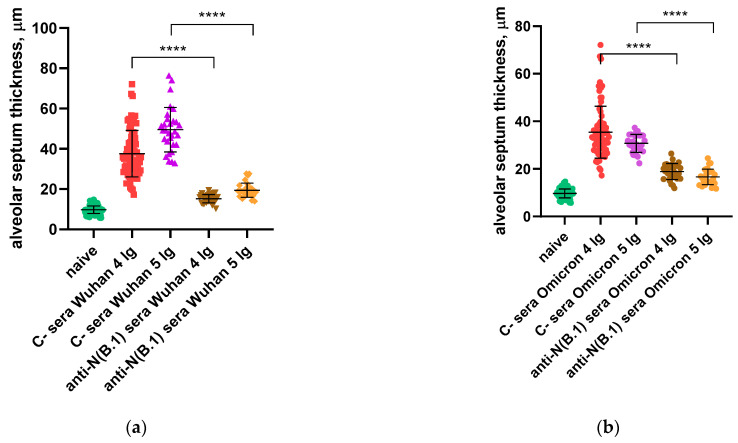
Analysis of changes in alveolar septum thickness in naïve or immunized hamsters as a result of challenge infection with Wuhan (B.1) (**a**) or Omicron (BA.2.86.1.1.18) (**b**) viruses. The calculations are performed at 200× magnification. ****—*p* < 0.0001.

**Figure 7 antibodies-14-00045-f007:**
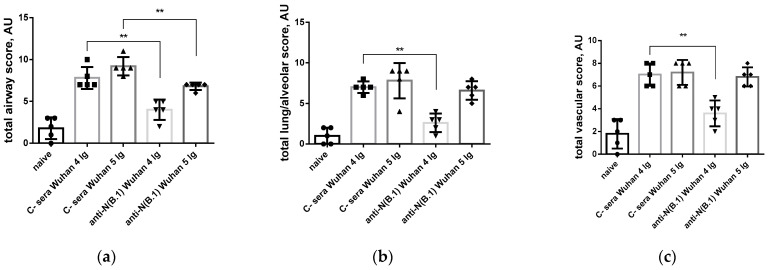
Morphometric lung data, obtained at 6^th^ day after SARS-CoV-2 infection with Wuhan (B.1) (**a**–**c**) or Omicron (BA.2.86.1.1.18) (**d**–**f**) viruses: (**a**,**d**) airway semi-quantitative assessment; (**b**,**e**) semi-quantitative assessment of alveolar lesions; (**c**,**f**) vessel bed semi-quantitative assessment. ** *p* < 0.01.

**Figure 8 antibodies-14-00045-f008:**
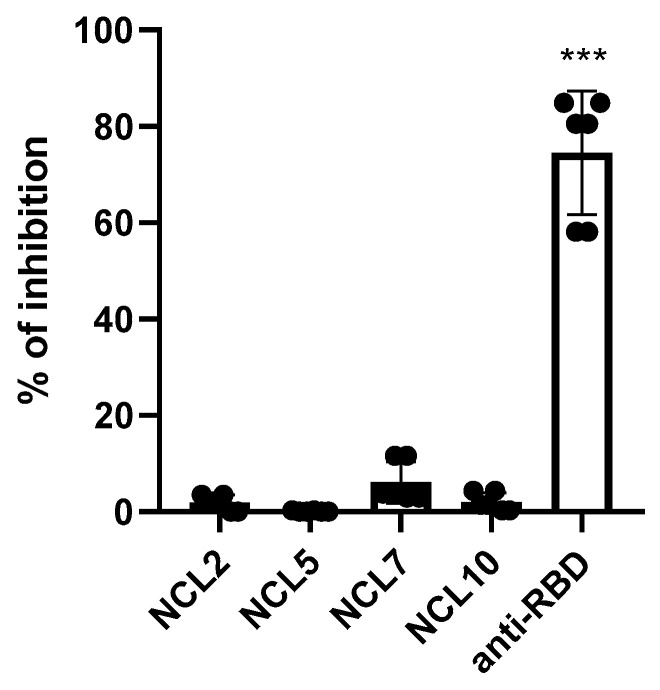
SARS-CoV-2 (B.1) neutralization assay by mouse monoclonal anti-N antibodies (NCLs) and anti-RBD antibody at MOI 0.01. *** *p* < 0.001.

**Figure 9 antibodies-14-00045-f009:**
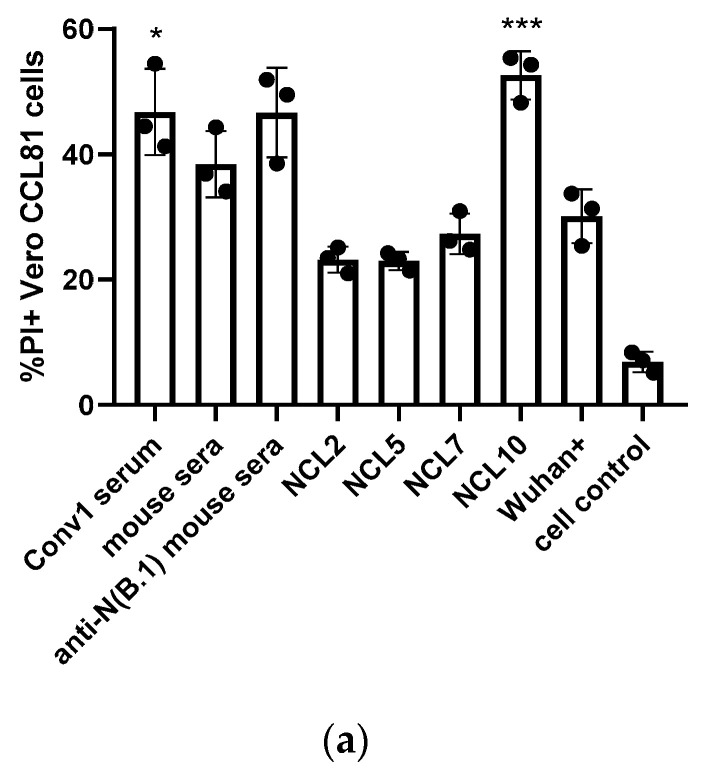
Analysis of the functional activity of anti-N antibodies–CDC induction: (**a**) CDC-promoting activity presented in graph bars; (**b**) representative CDC histograms (YoPro-FITC on the *X*-axis vs. propidium iodide (PI-PC5.5) on the *Y*-axis). * *p* < 0.05; *** *p* < 0.001.

**Figure 10 antibodies-14-00045-f010:**
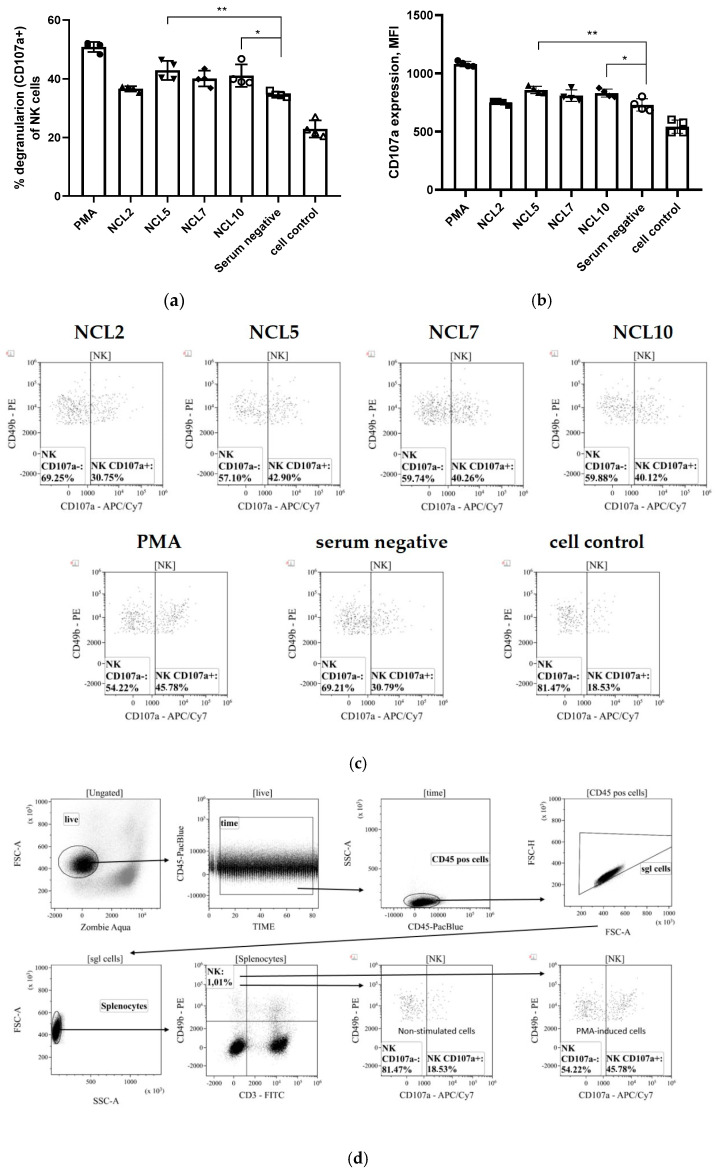
Analysis of the functional activity of anti-N antibodies–NK cells degranulation assay (ADCC evaluation): (**a**) NK degranulation-promoting activity presented in graph bars; (**b**) NK degranulation-promoting activity presented in MFI values; (**c**) representative ADCC histograms (CD107a-APC/Cy7 on the *X*-axis vs. CD49b-PE on the *Y*-axis); (**d**) gating strategy for NK cell degranulation assay. * *p* < 0.05; ** *p* < 0.01.

## Data Availability

The data presented in this study are available on request from the corresponding author.

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
