# Peer review of "Protective Potential and Functional Role of Antibodies Against SARS-CoV-2 Nucleocapsid Protein"

_2073-4468, 2025, doi:10.3390/antib14020045_

Round 1
Reviewer 1 Report
Comments and Suggestions for Authors
In this manuscript, the authors reported the protective roles that anti-N antibody played against COVID virus infection. Overall structure and writing is good, please see my comments and questions in below.
- Suggest to remove subtitles in the abstract to make a cohesive paragraph.
- Material section (2.1), please add the information of how to generate the sera, and how/where to get the recombinant N protein and anti-N antibodies.
- In section 2.2, please specify the total animal numbers, gender and explain how sample size and dosage amount were decided.
- In section 3.1, please reference Figure 1 in the paragraph. For Figure 1, 5 animals for 4 lg and 5 lg challenge respectively? Please specify in the text or figure caption.
- Please double check Figure 1b. It seems that 5lg lung viral titers also reached statistical significance.
- Figure 2, please explain why no impacts when challenged with 5 lg. Also explain and discuss Figure 2c in the paragraph.
- Line 199, unfinished sentence. Please proof read and complete the sentence.
- Figure 3, please add the explanation why there is no impact when challenged with Omicron virus.
- There is no explanation and discussion of Figure 4 in the context, please add.
- Line 207, add full name of H&E.
- Section 3.2 and Figure 7, what are the differences between the anti-N antibodies NCL 2, 5, 7 and 10.
- Figure 8a, explain what is the cell control. Why only NCL10 has impact while the other NCLs don’t?
- Figure 9, explain what is PMA and what are the conditions for cell control.
Author Response
Thank you very much for taking the time to review this manuscript and for a positive feedback. Please find the detailed responses below and the corresponding revisions/corrections highlighted in red in the re-submitted files.
In this manuscript, the authors reported the protective roles that anti-N antibody played against COVID virus infection. Overall structure and writing is good, please see my comments and questions in below.
Comments 1: Suggest to remove subtitles in the abstract to make a cohesive paragraph.
Response 1: Thank you for the suggestion. The subtitles have been removed from the Abstract to smooth the narrative flow.
Comments 2: Material section (2.1), please add the information of how to generate the sera, and how/where to get the recombinant N protein and anti-N antibodies.
Response 2: In according to the comment, we have added a description of the source and production of N(B.1) protein, anti-N(B.1) monoclonal antibodies and hamster sera in 2.1 subsection.
Comments 3: In section 2.2, please specify the total animal numbers, gender and explain how sample size and dosage amount were decided.
Response 3: We thank the reviewer for this important note. The details on hamster sample sizes, age, gender and immunization regimen have been added to the corresponding subsection.
Comments 4: In section 3.1, please reference Figure 1 in the paragraph. For Figure 1, 5 animals for 4 lg and 5 lg challenge respectively? Please specify in the text or figure caption.
Response 4: As suggested, the Figure 1 has been referenced in subsection 3.1, and sample sizes have been specified both in the figure caption and in 2.2 subsection.
Comments 5: Please double check Figure 1b. It seems that 5lg lung viral titers also reached statistical significance.
Response 5: Thank you for pointing this out. The text explaining the Figure 1, b has been modified as follows:
‘Surprisingly, anti-N (B.1) sera significantly reduced virus replication both in the upper and the lower respiratory tract if the challenge dose was 4 lg, while in case of 5 lg reduction in nasal viral titers compared to the mock group did not reach statistical significance (Figure 1, a, b).’
Comments 6: Figure 2, please explain why no impacts when challenged with 5 lg. Also explain and discuss Figure 2c in the paragraph.
Response 6: Thanks a lot for raising an interesting question. The possible explanation of anti-N(B.1) sera inefficiency at a dose of 5 lg and Figure 2c mentioning have been added to the corresponding paragraph.
Comments 7: Line 199, unfinished sentence. Please proof read and complete the sentence.
Response 7: We thank the reviewer for noting this. The interrupted sentence has been fixed.
Comments 8: Figure 3, please add the explanation why there is no impact when challenged with Omicron virus.
Response 8: We thank the reviewer for this important note. Our insights into the lack of a protective effect of serum against the Omicron variant have been introduced to the text.
Comments 9: There is no explanation and discussion of Figure 4 in the context, please add.
Response 9: We agree with this comment. Therefore, the description of the results given in Figure 4 has been added to the manuscript.
Comments 10: Line 207, add full name of H&E.
Response 10: Thanks for the fair comment. A deciphering has been added to the text.
Comments 11: Section 3.2 and Figure 7, what are the differences between the anti-N antibodies NCL 2, 5, 7 and 10.
Response 11: We agree with the feasibility of this clarification, and therefore have added a description of anti-N(B.1) properties in 3.2 subsection.
Comments 12: Figure 8a, explain what is the cell control. Why only NCL10 has impact while the other NCLs don’t?
Response 12: To clear the description of results, the cell control explanation as well as speculation on NCL10 impact has been added to 3.2.1 subsection.
Comments 13: Figure 9, explain what is PMA and what are the conditions for cell control.
Response 13: Many thanks for the comment. PMA and cell control description has been added to the 3.2.2 subsection.
Reviewer 2 Report
Comments and Suggestions for Authors
Rak et al. explore the protective capacity of antibodies against the conserved nucleocapsid (N) protein of SARS-CoV-2, a promising vaccine target due to spike protein variability. Using a passive immunization model in Syrian hamsters, they demonstrate that anti-N(B.1) antibodies offer protection against homologous virus challenge at low doses, but not against divergent Omicron strains. Functional assays reveal that these antibodies induce complement-dependent cytotoxicity (CDC) and antibody-dependent cellular cytotoxicity (ADCC). The findings highlight Fc-mediated innate immune activation as a key mechanism, though protection is strain-specific and limited. The authors also stress the need to evaluate autoimmune risks when designing N-based vaccines.
Comments for Authors
1. Flow Cytometry Duplication Concern:
The flow cytometry dot plots labeled “B.1 (Wuhan)+” and “NCL2” in Figure 8b appear visually indistinguishable, including near-identical apoptotic body distributions. This raises concerns of potential image duplication or mislabeling. Please review and clarify, and submit the original FCS files for validation.
2. In Vivo Protection & Histology Scoring:
The passive transfer experiments in Syrian hamsters are well-structured and demonstrate clear, strain-specific protection by anti-N(B.1) sera. However, the histopathology analysis would benefit from quantitative scoring of lung and vascular lesions to enhance statistical rigor, especially given the qualitative nature of much of Figure 5.
3. Imaging Consistency & Annotation:
The H&E and MSB-stained lung sections are visually compelling, supporting the protective effect of anti-N sera. Nonetheless, there are minor inconsistencies in image brightness/contrast across panels, which could affect interpretation. Please ensure uniform processing and include scale bars for accurate visual comparison.
4. Functional Assay Data Presentation (CDC/ADCC):
The inclusion of CDC and ADCC assays meaningfully strengthens the mechanistic basis of antibody function. However, the presentation would benefit from additional quantitative summaries (e.g., bar graphs or MFI values) to accompany dot plots and facilitate comparative analysis across antibody clones.
5. Discussion of Autoimmunity Risks:
The discussion appropriately acknowledges the potential risk of autoimmunity associated with anti-N responses. This point could be further expanded by referencing recent studies on N-protein cross-reactivity with host antigens and its possible implications in immunopathology, especially in the context of N-based vaccine strategies.
Author Response
Thank you for consideration of our manuscript. We appreciate the time and effort the reviewer has dedicated to providing insightful feedback on ways to strengthen our paper. Thus, with great pleasure we resubmit our article for further consideration. We have incorporated changes that reflect the detailed suggestions you have graciously provided. We also hope that our edits and responses we provide below satisfactorily address all the issues and concerns the reviewer has noted.
Rak et al. explore the protective capacity of antibodies against the conserved nucleocapsid (N) protein of SARS-CoV-2, a promising vaccine target due to spike protein variability. Using a passive immunization model in Syrian hamsters, they demonstrate that anti-N(B.1) antibodies offer protection against homologous virus challenge at low doses, but not against divergent Omicron strains. Functional assays reveal that these antibodies induce complement-dependent cytotoxicity (CDC) and antibody-dependent cellular cytotoxicity (ADCC). The findings highlight Fc-mediated innate immune activation as a key mechanism, though protection is strain-specific and limited. The authors also stress the need to evaluate autoimmune risks when designing N-based vaccines.
Comments 1: Flow Cytometry Duplication Concern:
The flow cytometry dot plots labeled “B.1 (Wuhan)+” and “NCL2” in Figure 8b appear visually indistinguishable, including near-identical apoptotic body distributions. This raises concerns of potential image duplication or mislabeling. Please review and clarify, and submit the original FCS files for validation.
Response 1: Allow us to assuage the reviewer's concerns. As suggested, we verified that the results containing in FCS files and panels included in Figure 8b were properly matched. The aforementioned similarity of flow cytometry dot plots (B.1 (Wuhan)+ vs NCL2) is also confirmed by the lack of statistically significant differences in % of cells in late apoptosis phase between wells containing only infected cells (Wuhan+) and NCL2/NCL5/NCL7 monoclonal antibodies, as shown in Figure 8, a. To further confirm this graphical representation and as requested by the reviewer, we have attached FCS files containing raw flow cytometry data.
Comments 2: In Vivo Protection & Histology Scoring:
The passive transfer experiments in Syrian hamsters are well-structured and demonstrate clear, strain-specific protection by anti-N(B.1) sera. However, the histopathology analysis would benefit from quantitative scoring of lung and vascular lesions to enhance statistical rigor, especially given the qualitative nature of much of Figure 5.
Response 2: Many thanks to the reviewer for the opportunity to make our manuscript even more informative. Besides the previously performed alveolar septum thickness calculation, we have quantitatively evaluated the lung tissue lesions by applying the scoring criteria separately to assess the extent of airway, pulmonary parenchyma, and vascular damage. The results are now shown in Figure 7.
Comments 3: Imaging Consistency & Annotation:
The H&E and MSB-stained lung sections are visually compelling, supporting the protective effect of anti-N sera. Nonetheless, there are minor inconsistencies in image brightness/contrast across panels, which could affect interpretation. Please ensure uniform processing and include scale bars for accurate visual comparison.
Response 3: We agree with this comment. Therefore, we have adjusted the brightness/contrast of the raw microphotographs equally and have included the scale bars.
Comments 4: Functional Assay Data Presentation (CDC/ADCC):
The inclusion of CDC and ADCC assays meaningfully strengthens the mechanistic basis of antibody function. However, the presentation would benefit from additional quantitative summaries (e.g., bar graphs or MFI values) to accompany dot plots and facilitate comparative analysis across antibody clones.
Response 4: We thank the reviewer for this note. The flow cytometry results of evaluation of ADCC-stimulating activity of anti-N antibodies have been described both by bar graph and by plot showing MFI values as suggested by the reviewer. They are now presented in Figure 10,a, b.
Comments 5: Discussion of Autoimmunity Risks:
The discussion appropriately acknowledges the potential risk of autoimmunity associated with anti-N responses. This point could be further expanded by referencing recent studies on N-protein cross-reactivity with host antigens and its possible implications in immunopathology, especially in the context of N-based vaccine strategies.
Response 5: Thank you very much for a positive feedback. As it was suggested by the reviewer, the Discussion section has been expanded by the overview of recent papers on possible autoreactivity and detrimental effects of anti-N antibodies.
Reviewer 3 Report
Comments and Suggestions for Authors
In the current manuscript titled "Protective potential and functional role of antibodies against SARS-CoV-2 nucleocapsid protein", the authors study the protective efficacy of antibodies raised against the Nucleocapsid (N) protein derived from the B.1 lineage. These antibodies were tested for their cross-protection against both the ancestral Wuhan strain and the Omicron variant of SARS-CoV-2. The authors focused on the N protein due to its high sequence conservation and abundant expression in SARS-CoV-2-infected host cells. Their findings demonstrate that passive transfer of anti-N (B.1) antibodies provided protection against the Wuhan strain, but only at low challenge doses, in a Syrian hamster model. No protection was observed against the Omicron variant, regardless of the challenge dose. The observed protection was mediated through Fc-dependent immune mechanisms, specifically complement activation and NK cell-mediated cytotoxicity, rather than direct virus neutralization. Based on these results, the authors concluded that anti-N (B.1) antibodies offer strain-specific protection likely via Fc-effector functions, and not through neutralization.
While the study is well executed and adds valuable insights to the field of SARS-CoV-2 vaccine development, several areas required clarification and revision. My comments are as follows:
Comments:
- Lines 188–193: Please describe Figure 2a first, followed by Figures 2b and 2c, in logical order. Currently, there is no mention of Figure 2b—what does it represent? Also, please clarify the term "AUC of weights". What does AUC stand for in this context, and how was it calculated?
- Figure 4: To maintain consistency with the layout in Figure 2, please switch Figure 4c with Figure 4b.
- Figure 5: The two images in the first row on the left appear to be repeated in the right-hand panels. Please indicate this in the figure legend or text. Alternatively, consider removing the duplicated images or replacing them with new representative naive H&E and MSB-stained sections.
- The term AUC is used in several figures without definition. Please provide the full form (e.g., Area Under the Curve) and explain how it was calculated and interpreted in each context.
- Line 111: Please provide the full form of MSB at first mention (e.g., Martius Scarlet Blue staining).
- Line 199 appears to be incomplete. Please revise or complete the sentence.
- Figure 9: Please include the gating strategy for the CD107a assay. A representative flow cytometry gating plot would strengthen the data presentation.
- Figure 8b: The 'X' and 'Y' axes are currently unclear. Please provide axis labels and a brief explanation of the plot in the figure legend.
- Figure 7: Please clarify what RBD refers to in this context (presumably the Receptor Binding Domain of the Spike protein) and explain why it was used in this experiment.
Author Response
We’d like to thank the reviewer for the positive feedback, valuable comments and opportunity to resubmit a revised copy of this manuscript. The point-by-point details of the revisions to the manuscript and our responses to the referee’s comments are listed below. We very much hope that the revised manuscript will be accepted for publication.
In the current manuscript titled "Protective potential and functional role of antibodies against SARS-CoV-2 nucleocapsid protein", the authors study the protective efficacy of antibodies raised against the Nucleocapsid (N) protein derived from the B.1 lineage. These antibodies were tested for their cross-protection against both the ancestral Wuhan strain and the Omicron variant of SARS-CoV-2. The authors focused on the N protein due to its high sequence conservation and abundant expression in SARS-CoV-2-infected host cells. Their findings demonstrate that passive transfer of anti-N (B.1) antibodies provided protection against the Wuhan strain, but only at low challenge doses, in a Syrian hamster model. No protection was observed against the Omicron variant, regardless of the challenge dose. The observed protection was mediated through Fc-dependent immune mechanisms, specifically complement activation and NK cell-mediated cytotoxicity, rather than direct virus neutralization. Based on these results, the authors concluded that anti-N (B.1) antibodies offer strain-specific protection likely via Fc-effector functions, and not through neutralization.
While the study is well executed and adds valuable insights to the field of SARS-CoV-2 vaccine development, several areas required clarification and revision. My comments are as follows:
Comments 1: Lines 188–193: Please describe Figure 2a first, followed by Figures 2b and 2c, in logical order. Currently, there is no mention of Figure 2b—what does it represent? Also, please clarify the term "AUC of weights". What does AUC stand for in this context, and how was it calculated?
Response 1: We thank the reviewer for the comment. AUC deciphering and calculation procedure have been added to the captions for Figures 2 and 4.
Comments 2: Figure 4: To maintain consistency with the layout in Figure 2, please switch Figure 4c with Figure 4b.
Response 2: As suggested by the reviewer, the panels of the Figure 4 have been swapped.
Comments 3: Figure 5: The two images in the first row on the left appear to be repeated in the right-hand panels. Please indicate this in the figure legend or text. Alternatively, consider removing the duplicated images or replacing them with new representative naive H&E and MSB-stained sections.
Response 3: We agree with this valuable comment. The use of the same microphotographs of lung tissue from naïve animals was noted in the Figure 5 caption.
Comments 4: The term AUC is used in several figures without definition. Please provide the full form (e.g., Area Under the Curve) and explain how it was calculated and interpreted in each context.
Response 4: We thank the reviewer for the comment. AUC deciphering and calculation procedure have been added to the captions for Figures 2 and 4.
Comments 5: Line 111: Please provide the full form of MSB at first mention (e.g., Martius Scarlet Blue staining).
Response 5: Thanks for the note; MSB deciphering has been added to the 2.2 subsection.
Comments 6: Line 199 appears to be incomplete. Please revise or complete the sentence.
Response 6: We thank the reviewer for noting this. The interrupted sentence has been fixed.
Comments 7: Figure 9: Please include the gating strategy for the CD107a assay. A representative flow cytometry gating plot would strengthen the data presentation.
Response 7: Thanks a lot for an opportunity to improve the representation of our results. We agree with this suggestion and therefore have included the gating strategy for the CD107a+ NK cells degranulation assay. It is now presented in Figure 10,d, and representative flow cytometry plots of NK cells degranulation are given in Figure 10,c.
Comments 8: Figure 8b: The 'X' and 'Y' axes are currently unclear. Please provide axis labels and a brief explanation of the plot in the figure legend.
Response 8: We agree with the comment. We have enlarged the letter designations and also included the description of axles in the figure caption.
Comments 9: Figure 7: Please clarify what RBD refers to in this context (presumably the Receptor Binding Domain of the Spike protein) and explain why it was used in this experiment.
Response 9: Thanks a lot for the note. RBD deciphering and anti-RBD antibody explanation has been added to the 3.2 subsection.
Round 2
Reviewer 2 Report
Comments and Suggestions for Authors
Authors have addressed my concerns and suggestions.